# Post-Processing and Surface Characterization of Additively Manufactured Stainless Steel 316L Lattice: Implications for BioMedical Use

**DOI:** 10.3390/ma14061376

**Published:** 2021-03-12

**Authors:** Alex Quok An Teo, Lina Yan, Akshay Chaudhari, Gavin Kane O’Neill

**Affiliations:** 1Department of Orthopaedic Surgery, National University Hospital Singapore, 5 Lower Kent Ridge Road, Singapore 119074, Singapore; alex_teo@nuhs.edu.sg (A.Q.A.T.), gavin_oneill@nuhs.edu.sg (G.K.O.); 2Department of Mechanical Engineering, National University of Singapore, 9 Engineering Drive 1, #07-08 Block EA, Singapore 117575, Singapore

**Keywords:** additive manufacturing, biomedical implants, post-processing, surface residue, stainless steel 316L

## Abstract

Additive manufacturing of stainless steel is becoming increasingly accessible, allowing for the customisation of structure and surface characteristics; there is little guidance for the post-processing of these metals. We carried out this study to ascertain the effects of various combinations of post-processing methods on the surface of an additively manufactured stainless steel 316L lattice. We also characterized the nature of residual surface particles found after these processes via energy-dispersive X-ray spectroscopy. Finally, we measured the surface roughness of the post-processing lattices via digital microscopy. The native lattices had a predictably high surface roughness from partially molten particles. Sandblasting effectively removed this but damaged the surface, introducing a peel-off layer, as well as leaving surface residue from the glass beads used. The addition of either abrasive polishing or electropolishing removed the peel-off layer but introduced other surface deficiencies making it more susceptible to corrosion. Finally, when electropolishing was performed after the above processes, there was a significant reduction in residual surface particles. The constitution of the particulate debris as well as the lattice surface roughness following each post-processing method varied, with potential implications for clinical use. The work provides a good base for future development of post-processing methods for additively manufactured stainless steel.

## 1. Introduction

Cellular metallic materials have enjoyed a boost in popularity in recent years, owing largely to increasing accessibility of additive manufacturing [1]. In attempting to recreate natures complex designs, scientists and engineers have sought to reap the benefits of these constructs; not only does their porous nature allow significant materials and cost-savings, they are more light-weight but retain good energy absorption and thermal insulation properties [2]. Where the production of smaller cellular metal structures was previously limited in complexity by existing manufacturing technology—subtractive laser cutting, chemical etching with photolithography, additive wire wielding and braiding techniques —current additive manufacturing technology allows increased structure complexity without a concurrent increase in cost.

One of the main potential applications for these materials is in the biomedical industry, within which manufacturing of customized implants with specific shape [3] or surface characteristics holds significant promise [4]. Orthopedic prostheses that allow bony ingrowth are commonly used in various joint replacement procedures, and it is believed that the increased strength of the bonds at the bone-metal interface improves the longevity of these procedures. Fully porous cellular Ti-6Al-4V structures have been produced [5,6], which allow bony in-growth when used as an orthopedic implant. This porosity has a secondary effect of reducing the Young’s modulus of the manufactured construct, thus helping minimize the problem of stress shielding [7].

Stainless steel 316L is a widely used austenitic stainless steel in a multitude of different industries including the biomedical industry. Its relatively low cost, ductility, high stiffness, and good corrosion resistance properties make it one of the most commonly used materials for fabrication of orthopedic implants [8]. Whether produced by conventional subtractive methods or via additive manufacturing, stainless steel 316L needs to undergo processing to reduce its high surface roughness and optimize its corrosion resistance prior to industrial use. Appropriate surface finishing is vital in the biomedical industry. A balance has to be struck between optimizing surface finish parameters to interact appropriately with the surrounding tissue while minimizing the amount of metal debris produced. Surface finishing of metals directly affects the amount of corrosion that occurs [9], leading to implant loosening and significantly reducing implant longevity. For instance, biomaterials comprised of titanium alloys are typically imbued with biocompatibility and corrosion resistance via a titanium oxide layer [10]. With stainless steel, passivation is applied. Depending on the surface treatment applied, however, these treatments could adversely affect the mechanical strength of the construct [11]. In bearing surfaces, the surface finishing of implants has a direct impact on their friction coefficient and therefore wear rates and lifespan [12,13]. In addition to its considerable impact on implant loosening from the generation of debris, surface modification of implants also influences the release of metal ions from cyclic formation of surface oxides and oxyhydroxides, and there is concern regarding the long-term health effects of these metallic ions [14,15].

Owing to its relatively recent emergence, much of the focus has been on the ideal construct design and the printability of stainless steel lattices, with relatively less attention paid to surface finishing, which is a major hurdle that needs to be overcome prior to more widespread use [16,17]. Standards for processing of conventional stainless steel are well-defined and widely accepted. Standards ASTM F86, A380, A967, and B600 [18] describe standard procedures for cleaning, descaling, and passivation of biomedical implants. Various mechanical and electrochemical surface treatments have been introduced to remove the residual iron particles, ceramic media, and other foreign particles that are introduced at each of the forming, machining, tumbling, and bead blasting processes. Standard ASTM F2791 [18] describes common methods for assessment of surface finishing characteristics but does not apply to porous structures with pore dimensions exceeding 0.05 µm.

In contrast, there are currently no benchmarks nor standards to guide additive manufacturing and post-processing of devices for orthopedic surgery. Current metal additive manufacturing processes are primarily powder-based, which results in relatively rough metallic cellular lattices produced [19]. For example, layered surfaces with significant powder attachments were seen on scanning electron micrograms (SEM) of both stainless steel [20] and titanium microlattices [5,17] produced by SLM, which has a bearing on the ultimate tensile strength of the materials produced. Furthermore, compared to its wrought counterpart, additively manufactured stainless steel is known to contain microstructural defects [21] inherent to the production process. Both the surface roughness and the increased porosity are thought to contribute to its reduced fatigue strength [22]. Increased porosity and surface roughness of additively manufactured titanium also increases bacterial colonization [23], which can be logically extrapolated to an increased risk of implant infection if not adequately addressed via surface treatments. Post-processing of additively manufactured materials is a rapidly developing area of study in attempts to improve the performance and reliability of these materials. Various post-processing methods for additively manufactured stainless steel 316L have been studied in recent years [24,25,26]; however, the use of these methods is generally limited to simple shapes. Post-processing of more complex parts is less well-described.

The optimal post-processing protocol has yet to be established, with each additional process adding to the manufacturing cost. Additively manufactured metals including stainless steel are known to possess a dissimilar microstructure owing to differences in the production process—differences exist both as a whole as well as between the various production methods for additive manufacturing [27]. These translate to altered and less predictable surface and biomechanical properties [28]. Properties such as corrosion resistance and biocompatibility hinge upon the surface finishing and have the potential to significantly affect the performance of the additively manufactured product. This is crucial in biomedical applications where printed devices are implanted and not easily removed, modified or replaced once in-vivo. Correlating surface parameters to device performance is challenging, as there are as yet no universally accepted characterization parameters to define these additively manufactured devices nor thresholds for safe use. There has been some suggestion that those used for conventional wrought stainless steel may be unfit for use, and that novel parameters for additively manufactured surfaces may be required [29]. Brown et al. have highlighted importance of the multiscale analysis for the powder-based additive manufacturing [29]. Various methods are being used for the characterization powder-based 3D printing methods such as contact [30] and non-contact profilometry [31,32,33] and micro-X-ray computer tomography (microCT) [34,35,36]. There remain limitations to these methods—contact-type measurements for instance are only suitable for flat and shallow surfaces, whereas optical microscopy fails to capture the profiles of internal surfaces. MicroCT on the other hand is able to capture the internal surfaces but requires complex data processing to fully capture surface features [37].

We anticipate that additive manufacturing of stainless steel will continue along its trajectory of burgeoning popularity, and yet relatively little has been published about optimizing the surface of the manufactured biomedical product. This work was thus carried out to further our collective understanding of the impact of various post-processing techniques on additively manufactured stainless-steel lattices for surgical implants. We have investigated the surface features of an additively manufactured stainless steel 316L lattice via confocal and scanning electron microscopy and have further examined the chemical composition of the residual debris and foreign particles using energy dispersive X-ray analysis, after various combinations of post-processing methods.

## 2. Materials and Methods

### 2.1. Lattice Design

An octahedral lattice was designed using Materialise 3-matic software (Materialise, Leuven, Belgium) with extra pillar strands added to the surface plane (Figure 1) to reinforce the bending strength for load-bearing orthopedic applications. Trusses were oriented at 45° and 55° in the 3D system, with the radius set at 0.3 mm in a lattice unit of 4.25 × 4.25 × 3 mm^3^. This classic lattice design provides uniform attachment of metal powders on trusses fabricated using the powder-based SLM process. This design achieves sufficient construct stiffness to resist deformation while reducing the Young’s modulus compared to conventional stainless steel to reduce the problem of stress shielding (Singapore patent application number 10201902254Y).

### 2.2. Printing Parameters

The lattices were manufactured via SLM using stainless steel 316L powder provided by Renishaw (Renishaw plc, Wotton-under-Edge, UK). Recycled powder (less than 10 reuses) was used in the printing, as the elemental composition of the powder feedstock is known not to change significantly even after 30 cycles of reuse in SLM [38]. The nominal powder composition is shown in Table 1. A Renishaw AM 400 metal printer (Renishaw plc, Wotton-under-Edge, UK) was used with printing parameters shown in Table 2. Pulsed laser exposure was used during printing process, which induces a relatively narrow melt pool thus reducing the thermal stress on the thin structures produced [39].

### 2.3. Post-Processing of Lattice

Post-processing of metal 3D-printed parts is required prior to use, at a minimum to remove excess powders and support structures but additionally to achieve desirable surface and mechanical properties, with numerous different techniques described for use. Sandblasting is commonly used due to its ability to efficiently remove loosely bound particles [40]. Other techniques, such as shape adaptive grinding [41], abrasive flow finishing [42], and magnetic abrasive polishing [26], have also been used for the post-processing of internal and external surfaces. Many of these techniques inadequately access internal surfaces, and are thus inappropriate for use with lattice structures. Internal surfaces can only be accessed by loose abrasive media. Abrasive polishing was thus selected as another post-processing method for this study, as the loose abrasive media is able to access the internal lattice surfaces. Finally, electropolishing is employed as a final finishing process to improve surface quality and supplement unwanted material removal.

Firstl standard stress release annealing was conducted at 475 °C for four hours. Second, further post-processing of the lattice structure was carried out using various combinations of one to three methods—sandblasting, abrasive polishing and electropolishing—with each method applied separately as detailed below. The experimental plan is shown in Figure 2. The specimens were cleaned in ultrasonic alcohol bath after completion of each post-processing step.

#### 2.3.1. Sandblasting

After lattice production, standard sandblasting was applied to all the specimens to remove any loosely bound or partially molten powder particles (Experiment 1A). Sandblasting was carried out using a commercially available machine (Peenmatic 750S, Swiss Instruments Limited, Mississauga, ON, Canada) with a nozzle diameter of 5 mm. The average distance between the sample and the workpiece was kept at 100–120 mm. Pressures of between 60 psi and 80 psi, and glass beads with diameters of 200–300 µm were used. The glass beads were provided by the equipment manufacturer and were produced via large-scale fabrication, with high levels of impurities within each batch.

#### 2.3.2. Abrasive Polishing

Abrasive flow polishing is widely used for processing complex cavity parts. A pressurized slurry flow is applied into a micro hole with a turbulent flow regime, a typical setup is shown in Figure 3 [43]. In this study, the polishing media comprised of silicon carbide (SiC), aluminum oxide (Al_2_O_3_), boron carbide (B_4_C), and steel grit with 3%, 4%, 3%, and 90% weight percentages, respectively. Abrasive polishing is an established technology for polishing of stainless steel and ceramic bores [44].

Abrasive polishing was performed after sandblasting on the specimens in experiment 2A. A custom-built vibration-assisted polishing device was used (Figure 3). The setup consists of a pneumatic piston with a fixed vibration frequency of 30 Hz. The polishing chamber was fabricated using acrylic sheets and contains a cavity to hold a workpiece atop a fixture plate; this is attached to the piston via a link. The chamber is then filled with an abrasive slurry and the cover fastened using a clamp. As the piston vibrates, the particles inside the chamber remove excess material from the surface of lattice. In this study, the steel grit in the abrasive slurry has an average particle size of 150 µm whereas the average size of the other abrasive particles was below 10 µm. The steel grit was again produced via large-scale fabrication, and therefore contained a high level of impurities as expected.

#### 2.3.3. Electropolishing

In the electropolishing (EP) process, lattice samples were positively charged (anodic), and metal ions are thus removed from the samples with application of the current. Electropolishing was performed as the final step in the post-processing sequence on the sandblasted samples with or without abrasive polishing (experiments 1B and 2B). For the as-printed sample, the metal powders were partially attached to the lattice (Figure 4). The EP process results in simultaneous dissolution of these partially attached powder particles together with the lattice surface, and is thus ineffective in removing the attached powders on the as-printed samples. For the sandblasted and abrasive polished samples, the resultant debris on the lattice surface was effectively removed by EP.

In this study, electropolishing was carried out via a commercially available setup with reference to standard ASTM A380 [18] for surgically implantable stainless steel devices. The setup utilized a chemical solution of H_3_PO_4_ 90% and H_2_SO_4_ at 45–50 °C for 10–30 min. The workpiece was rinsed with deionized water after completion of the electropolishing process to remove any residual solution, and allowed to dry before use.

### 2.4. Digital Microscopy

The surface topography of the lattices was examined using a LEXT OLS5000-SAF (Olympus Corporation, Tokyo, Japan) confocal microscope at 20 × magnification. The measurements were performed at the joints and on the side planar surface for comparison. Measurement areas of 1221.5 × 1218.5 µm^2^ were captured using the stitching function. The measured data was processed using LEXT analysis software (Olympus Corporation, Tokyo, Japan) to obtain height and hybrid parameters according to ISO 25178 [45]. Three joints were tested for each lattice form, and the average values were calculated to form a representative value for that lattice.

### 2.5. Scanning Electron Microscopy

The surface features of the lattices were further examined using a field emission scanning electron microscope (FESEM) Hitachi S-4300 (Hitachi, Tokyo, Japan) under secondary electron imaging mode. To further investigate the chemical composition of the residual particles from each post-processing method, an energy dispersive X-ray detector—“X-act” (Oxford Instruments plc, Abingdon, UK)—attached to the FESEM was used for energy-dispersive X-ray (EDX) analysis.

## 3. Results and Discussion

### 3.1. As Printed

The melt material slips into layer below during the SLM process—a commonly observed phenomenon with additively manufactured stainless steel [27]—and as a result rough edges can be observed at the under surface of the lattice joints. Moreover, clusters of unmolten particles can be observed at the joints as shown in Figure 4B. The attached powders are approximately 20–30 µm in size (Figure 4), consistent with the powder size range in table 2. At the joints, the measured Sa and Sz values were 15 µm and 55 µm, respectively.

### 3.2. Sandblasting

The sequential deposition of layers during the SLM process results in partially molten metal powders attaching to the surface of the lattice as demonstrated in Figure 5. Sandblasting can effectively remove these unwanted surface materials including unmolten and loosely bound particles to improve its surface characteristics and is thus commonly used for post-processing of additively manufactured components. At the same time, however, local plastic deformation occurs from the impact, which ultimately affects surface roughness. The extent to which this occurs depends largely on parameters used, with both increased pressure and increased blasting duration leading to increased surface roughness [46]. The selection of parameters for sandblasting—which also include particle size and material in addition to the above-mentioned pressure and duration—is generally based on the operator’s experience. Absolute control over the pressure and blasting duration is frequently not possible due to manual control of the nozzle. Selection of abrasive particle size and material is thus important. Selection of higher abrasive size may lead to breaking of links or excessive material removal. The use of metal beads results in a more abrasive process, while glass beads may shatter and create sharp edges which result in unwanted damage to the blasted surface. Sandblasting of the lattice structure results in the formation of a peel-off layer on its surface (Figure 6). As the high velocity abrasive particles impact the link at an angle, the material is ploughed in the direction of particle velocity. There is resultant loss of material and, more significantly, plastic deformation. In addition, abrasive particles used for blasting may also get lodged within this peel-off layer and become difficult to remove. The measured Sa and Sz were 5 µm and 37 µm, showing a significant reduction in the surface roughness compared to the as-printed lattice.

A representative scanning electron micrograph of the lattice surface after sandblasting is shown in Figure 7. Compared to the surface of the as-printed lattice in Figure 4, there is a significantly reduced number of spherical particles; this explains the significant reduction in surface roughness seen on digital microscopy. The presence of a peel-off layer, however, suggests that the sandblasting process resulted in severe lattice surface damage. Furthermore, irregular particles were seen on the surface of the lattice. EDX analysis of these particles is shown in Figure 8. Compared to the base composition of the stainless steel 316L lattice shown in Table 1, the additional foreign elements of B, Bi and Ca found in the debris suggest that they indeed arose from the glass beads of B_2_O_3_ or B_4_C, Bi_2_O_3_ and CaO used in the sandblasting process. The weight percentages of these elements were calculated from the intensity of the peaks (Figure 8); calcium was not counted due to the small amount present. Heiden et al. [38] observed that for powders in the highly reused state (>30 reuses), there was a slight increase in Cr, Ni, Mn, P, S, N, and a slight decrease in Fe, Mo, and Si. In our study, the significantly higher weight percentage of Mg compared to the base stainless steel 316L (<2%) again suggests magnesium oxide debris deposition from the glass beads rather than simply from powder reuse.

### 3.3. Abrasive Polishing

The subsequent use of abrasive polishing following sandblasting (experiment 2A) resulted in removal of the above-mentioned peel-off layer (Figure 9 and Figure 10). The measured Sa value was 5 µm which is comparable to the value from experiment 1A. The Sz value however reached 53 µm which is comparable to the as-printed condition. The increase in the Sz is likely attributable to the surface pits introduced in the abrasive polishing process. Moreover, residual particles with diameters of between 2 µm and 5 µm were observed. Figure 11 shows the EDX analysis of these residual particles–there are distinct peaks of Si and Ca but no Bi, suggesting a change in the composition of the surface debris after abrasive polishing. Furthermore, the weight percentage of Si was again much higher than the base levels of Si in highly reused stainless steel 316L powder (<1%) [38,47,48]. The EDX mapping in Figure 11 further reveals a relative excess of Si and a deficiency of other elements in the region of the spherical particle, confirming that the main composition of the particle is SiO_2_.

This combination of sandblasting and abrasive polishing produces surface pits leading to increased Sz and also introduces foreign particles that are different from that produced by the sandblasting process alone. Electropolishing was thus introduced as an additional step to further reduce the amount of surface debris and residual particles.

### 3.4. Electropolishing

When electropolishing followed sandblasting (experiment 1B), the abovementioned peel-off layer was removed. As a result, the Sa was reduced from 5 µm to 3 µm, comparing experiments 1A and 1B. There were however microcracks that developed as a result of this layer removal in addition to material dissolution during the electropolishing process (Figure 12). In addition, these cracks likely make the lattice susceptible to pitting corrosion, which could further form internal cavities or holes. Figure 13 shows the enlarged internal cavities following electropolishing. This explains the increase in Sz from 37 µm to 40 µm in 1A and 1B. Despite removal of the peel-off layer and some residual debris, some micron level residual particles introduced in the sandblasting process remained, as shown in Figure 14A. A representative EDX analysis of these particles is shown in Figure 14C—the spectrum peaks are similar to those seen with the sandblasted surface alone (experiment 1A), but without the small peaks of Ca and Mg. This can be attributed to removal of residual glass beads that were entrapped within the peel-off layer.

Finally, when sandblasting, abrasive polishing and electropolishing were performed sequentially (experiment 2B), Sa was increased to 12 µm, which was higher than that in experiment 2A. This may have been caused by pitting corrosion occurring within the deep cracks that were created during the sandblasting process and were further enlarged during abrasive polishing. Sz, however, remained relatively stable at 51 µm.

Additionally, there were also noticeably fewer foreign particles remaining on the surface of the lattice (Figure 15). These particles were again analyzed via EDX, with the results shown in Figure 16; high weight percentages of B, Bi, Si, and Al again suggest residual glass bead particles composed of B_4_C, B_2_O_3_, and Bi_2_O_3_ from the sandblasting process, as well as SiO_2_ and Al_2_O_3_ from the abrasive polishing process (Figure 16). Mapping of the particulate matter confirmed Al to be the predominant element (Figure 17). Electropolishing successfully reduces the amount of loosely attached debris on the lattice surface, but at the cost of introducing microcracks from material removal, as well as the widening of internal cavities from pitting corrosion.

### 3.5. Residual Particles

While the combination of all three processes (experiment 2B) reduces the overall volume of particulate debris, some particles yet remain. EDX analysis of these particles reaffirms them to have been introduced by the various processing methods. The significances of these foreign particles and micro-scale debris are uncertain. If used for in-vivo implantation, they could theoretically incite both local and systemic effects. Ions and nanoparticles from orthopedic implants are known to be gradually released from the surface into the bloodstream [49,50]. Despite the inert nature of the base constitutive materials, there is some evidence to suggest that at least some of these ions and nanoparticles have deleterious effects at a cellular level [50,51,52]. Whether or not these eventually amount to any clinical significance long term remains to be seen [53,54,55], but mindful vigilance at the very least is required with future work around these in-vivo applications.

### 3.6. Surface Topography

The surface topography measurements are shown in Figure 18. The measured height parameters for both the joint and side planar surfaces are shown in Figure 19, with higher values indicating more loosely bound or partially molten particles. Measured skewness (Ssk) and kurtosis (Sku) values indicate a high degree of skewness due to peaks on the surface (Figure 20). These parameters are believed to be insufficient to describe the surface [37], and thus hybrid parameters root mean square gradient (Sdq) and developed interfacial area ratio (Sdr) were measured (Figure 20 and Figure 21).

Townsend et al. [34] reported that ISO 25178-2 [45] parameters such as peak material volume (Vmp), Sdr, reduced peak height (Spk), and Ssk are sensitive to vibro-finishing processes. Sdr provides a good indication of surface complexity and decreases after sequential post-processing as the peaks are removed from the surface. Interestingly in our study, Sdr values for the side planar surfaces were higher than that for joint surfaces. This can be attributed to the higher exposure to sandblasting and material removal at the microcracks during electropolishing. The combination of sandblasting and abrasive polishing results in a reduction of both conventional height and hybrid parameters. Electropolishing however, results in material removal and formation of surface valleys and pits; an increase in surface texture values was thus observed. A summary of the effects of the various post-processing treatments is shown in Figure 22. Though this analysis was limited to exposed lattice surfaces, it provides an overview of the effect of various post-processing methods on the surface topography of an additively manufactured stainless-steel lattice. Future work could involve use of MicroCT to evaluate the internal surfaces of the lattice.

## 4. Conclusions

Additive manufacturing of stainless steel 316L is rapidly improving, with a myriad of potential applications. In its current state, however, several key hurdles pertaining to the surface quality of the final product need to be overcome prior to more widespread use. Post-processing with sandblasting, abrasive polishing and then electropolishing introduces microcracks and leaves foreign particles on the surface, leading to relatively high surface roughness. Further work is required for correlation of these to material performance, particularly corrosion resistance.

Residual porosity and surface roughness in additively manufactured stainless steel are known hurdles, and are thought to contribute to its reduced fatigue strength [56]. A comprehensive review of post-processing techniques is beyond the scope of this study, but numerous different techniques have been described to improve the surface qualities of additively manufactured stainless steel. Elangeswaran et al. [57] found that surface machining but not heat treatment improved the surface characteristics of stainless steel 316L produced by SLM and resulted in improved fatigue resistance. These were, however, performed on a non-lattice structure; effective surface machining of lattices is not possible with current technology. Its effect on corrosion needs to be studied further. Kaynak and Kitay [25] were similarly able to reduce the surface roughness of additively manufactured stainless steel 316L rods via finish machining, drag finishing, and vibratory surface finishing but conceded that not all of these methods would be suitable for complex shapes.

Other groups have had more success with thermally based treatments. Obeidi et al. [58] successfully reduced the average surface roughness of stainless steel cylinders via laser polishing. This method has the benefit of avoiding the residual debris discussed above, but again may be difficult to apply to more complex shapes. Cost may also be prohibitive for larger samples. These will also need to be correlated with its mechanical strength and corrosion resistance. Chemical-based methods such as low-temperature plasma nitriding [59] have been shown to successfully reduce surface porosity with improved wear and corrosion resistance; such methods theoretically address all surfaces of the additively manufactured part and could be used for complex shapes including lattices, but their combined effect with the other processes particularly with regard to the surface particles needs to be ascertained.

The successes of numerous groups in reducing surface roughness, surface porosity and improving wear and corrosion characteristics of additively manufactured stainless steel 316L through various means is encouraging. The surface particles reported in this study are novel, and appropriate handling of them and their implications have not yet been uncovered. For biomedical applications, the optimal combination of post-processing methods would need to be determined, and it must be shown that the reduced surface porosity and roughness lead to improved corrosion and wear resistance without adversely affecting mechanical strength; they must be shown to at least be equivalent to characteristics of current wrought stainless-steel parts. Moreover, we must be certain of the inertness of the residual surface debris. When used for biomedical applications in-vivo, there is potential for the residual surface particles and micro-scale debris to leach into the bloodstream and induce local or systemic reactions. The potential systemic effects of these particles are a critical issue that needs to be addressed prior to routine implantation.

Additionally, refinement of all the surface processing techniques described would likely further improve the surface qualities of the additively manufactured metals. Automation of the sandblasting process may minimize the amount of deformation and loss of material that occurs. Similarly, with both abrasive polishing and electropolishing, fine-tuning of the parameters used would likely go a long way in achieving the desired smooth surface without excessively damaging the native material surface. Automated variations of the frequency and duration of polishing will need to be explored to determine the optimal settings for this. This, in concert with differing slurry compositions for abrasive polishing is an area for future work.

One of the main purported benefits of additively manufacturing metal is the rapidity of the production process, and with current surface finishing technology it is unlikely that undergoing the three separate processes described above is going to be cost- or time-effective, particularly if they each require human labor. Future large scale additive manufacturing is likely to rely on a single-stage post-processing technique.

## Figures and Tables

**Figure 1 materials-14-01376-f001:**
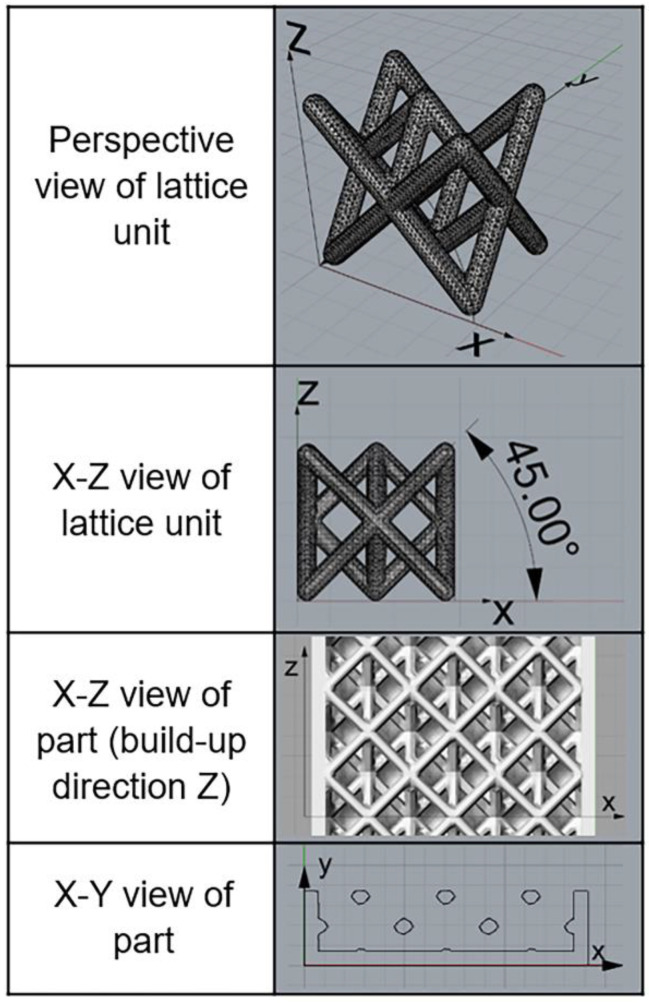
Lattice design and the 3D spatial arrangement of the selective laser melting process.

**Figure 2 materials-14-01376-f002:**
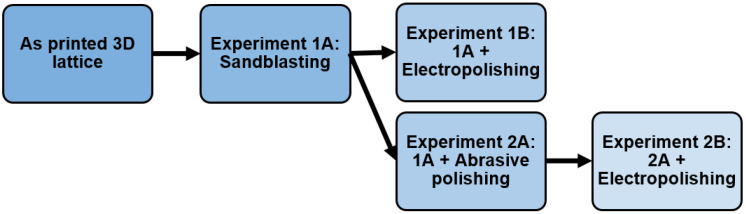
Sequence of post-processing techniques employed in the various experimental groups.

**Figure 3 materials-14-01376-f003:**
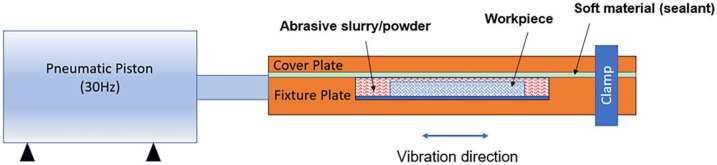
Abrasive polishing device setup.

**Figure 4 materials-14-01376-f004:**
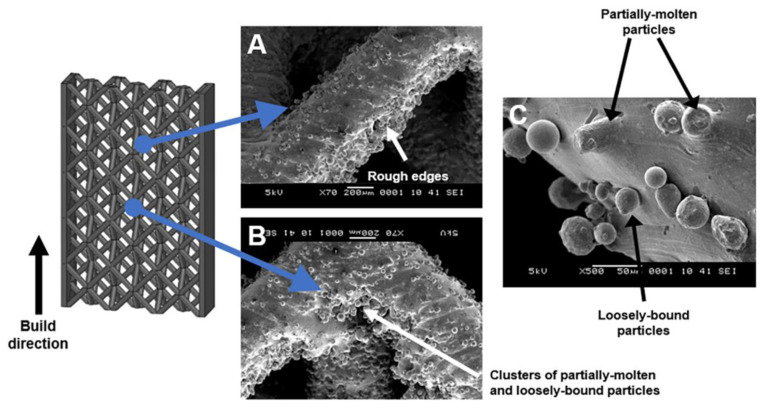
Scanning electron micrographs of the as-printed sample. (**A**) Rough edges on the under surface of the lattice. (**B**) Clusters of partially molten material at the joint. (**C**) Higher magnification image of the partially molten and loosely bound particles.

**Figure 5 materials-14-01376-f005:**
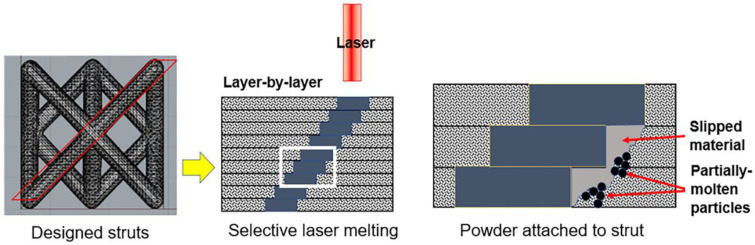
Diagrammatic representation of the selective laser melting process.

**Figure 6 materials-14-01376-f006:**
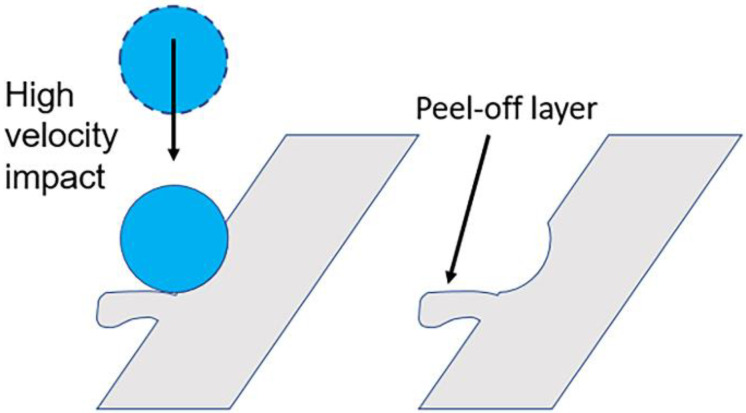
Mechanism of peel-off layer formation.

**Figure 7 materials-14-01376-f007:**
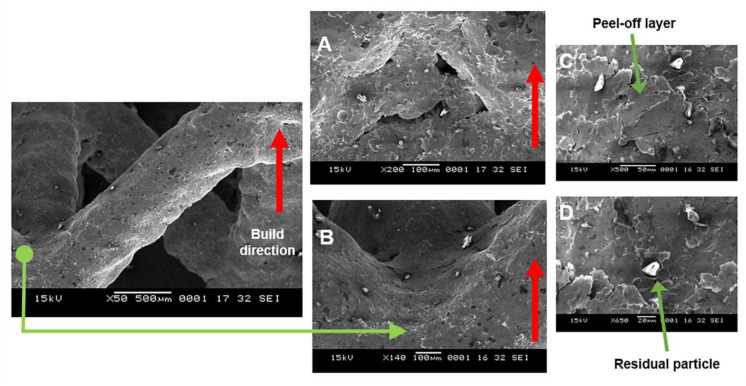
Scanning electron micrographs of lattice structure following sandblasting (experiment 1A). (**A**) Cavities at the joints. (**B**) Cavity free upper surface of joint. (**C**) Peel-off layer. (**D**) Residual irregular particles from the sandblasting process.

**Figure 8 materials-14-01376-f008:**
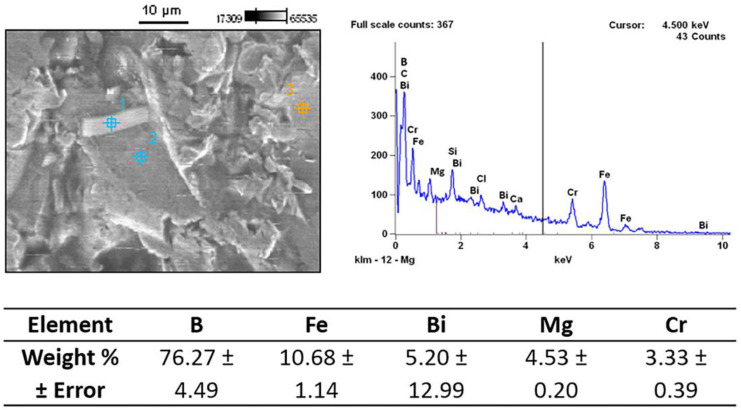
Energy dispersive X-ray (EDX) analysis of the residual surface debris after the sandblasting process. Three different regions were analyzed (**1**, **2** and **3**), and the energy dispersion spectrum and corresponding average weight percentages of the constitutive elements from the EDX analysis are shown.

**Figure 9 materials-14-01376-f009:**
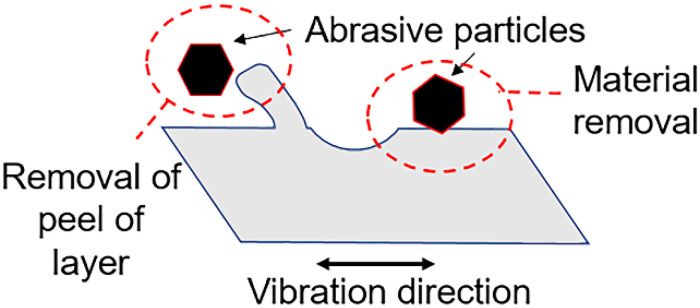
Removal of the peel-off layer via abrasive polishing.

**Figure 10 materials-14-01376-f010:**
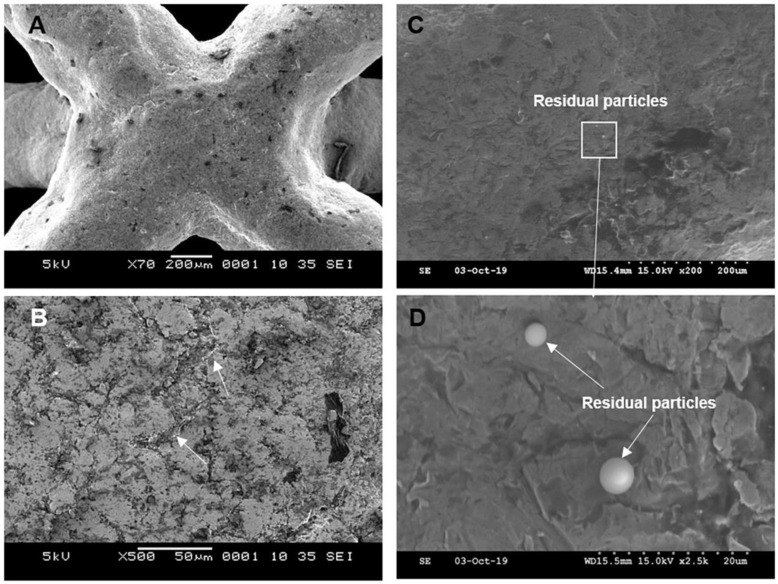
Scanning electron micrographs of lattice structure following sandblasting and abrasive polishing (experiment 2A). (**A**) Lattice structure. (**B**) Arrows indicating areas of peel-off layer removal/smoothening. (**C**) and (**D**) Residual particles at the end of the experiment.

**Figure 11 materials-14-01376-f011:**
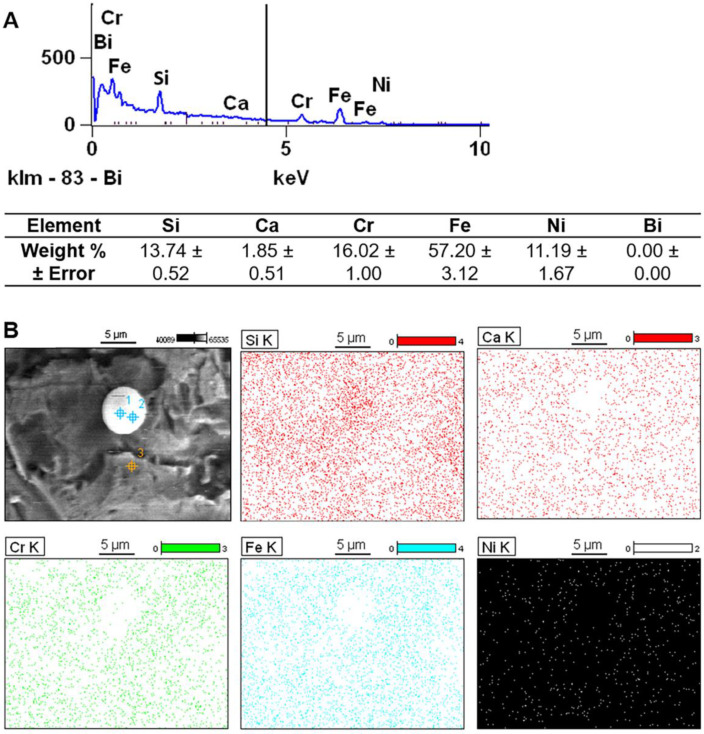
(**A**) Energy dispersive X-ray (EDX) analysis of the residual particles at locations 1 and 2 (location 3 was measured for reference). (**B**) X-ray mapping showing the elemental distribution in the region of the spherical particle.

**Figure 12 materials-14-01376-f012:**
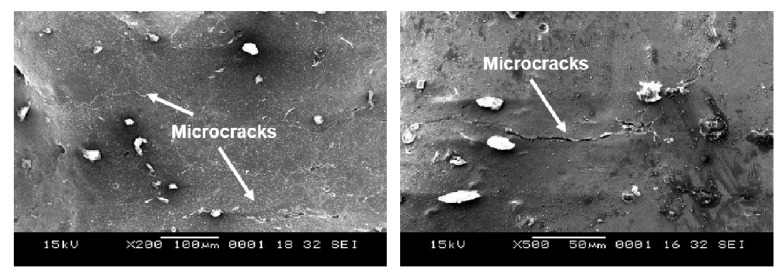
Formation of microcracks with sandblasting followed by electropolishing (experiment 1B).

**Figure 13 materials-14-01376-f013:**
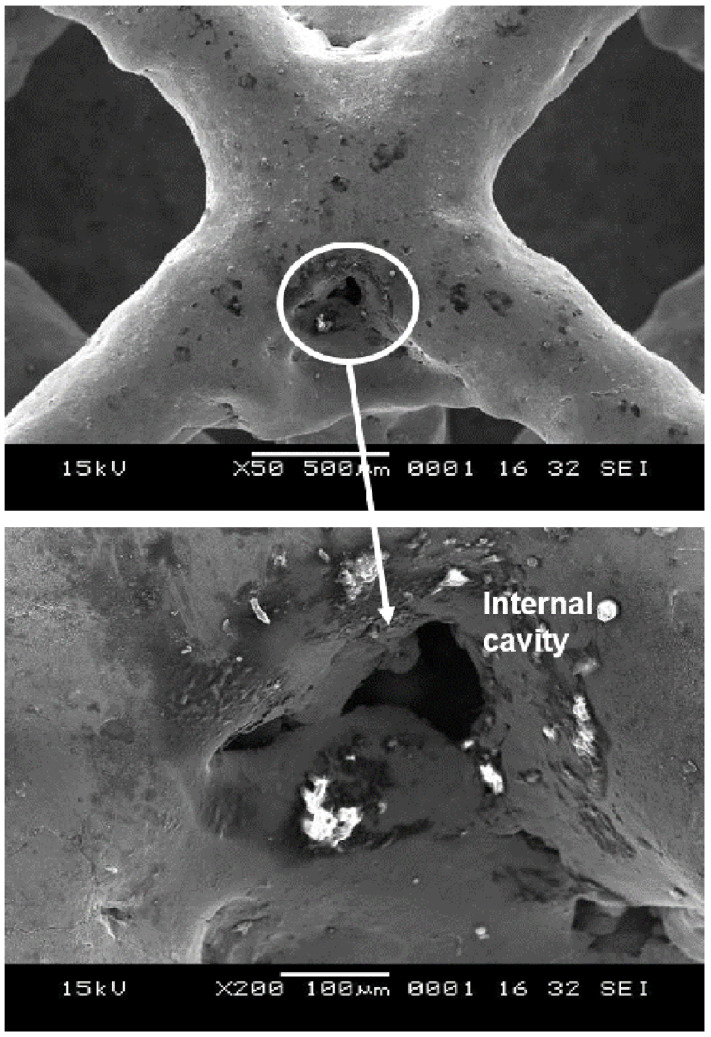
Amplification of internal cavities with sandblasting followed by electropolishing (experiment 1B).

**Figure 14 materials-14-01376-f014:**
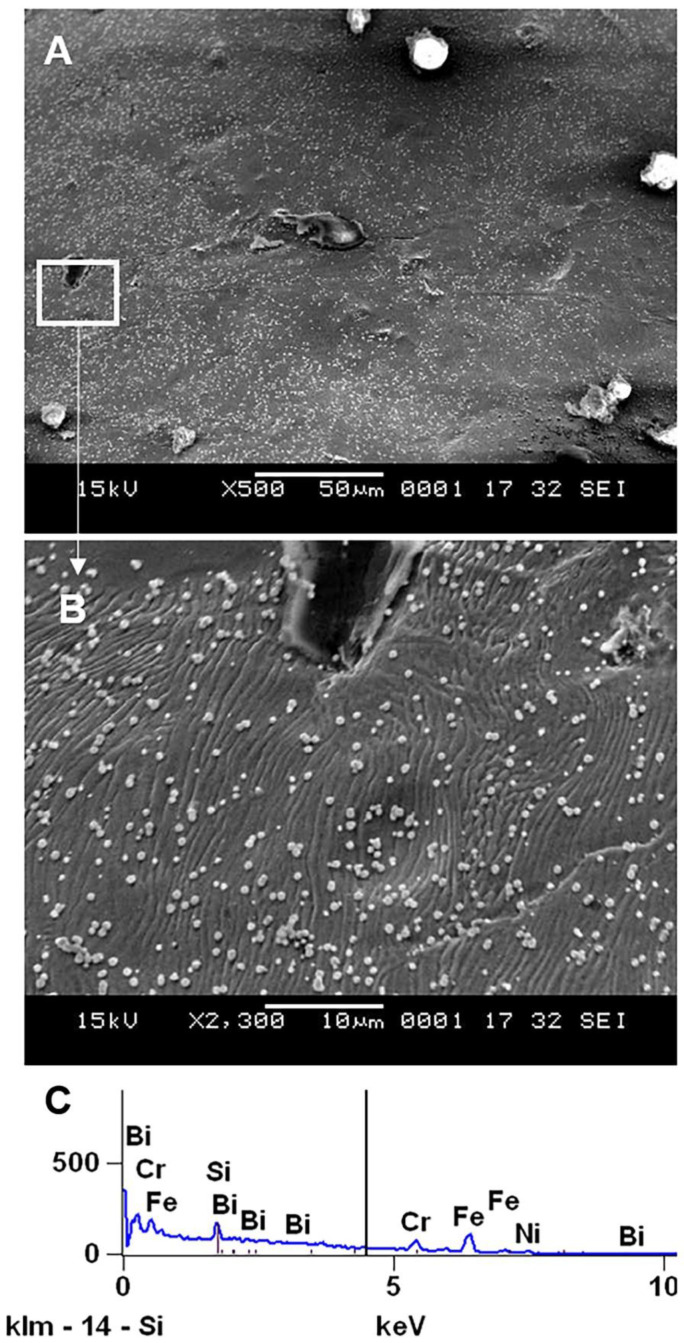
(**A**) Residual foreign particles with sandblasting followed by electropolishing (experiment 1B). (**B**) Further magnified rectangular region indicated on (**A**). (**C**) EDX analysis of these particles.

**Figure 15 materials-14-01376-f015:**
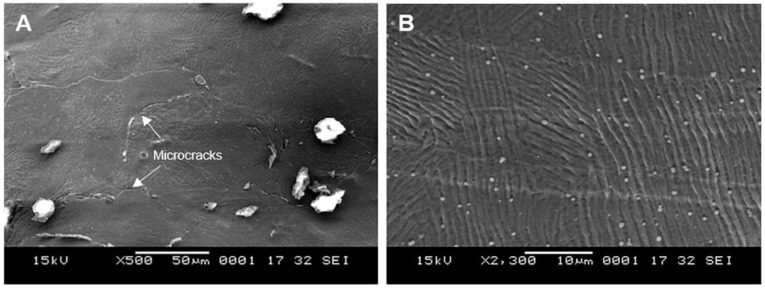
Scanning electron micrographs of the lattice following sandblasting, abrasive polishing and electropolishing (experiment 2B). Compared to experiment 1B, there were (**A**) shallower microcracks formed. and (**B**) fewer residual particles.

**Figure 16 materials-14-01376-f016:**
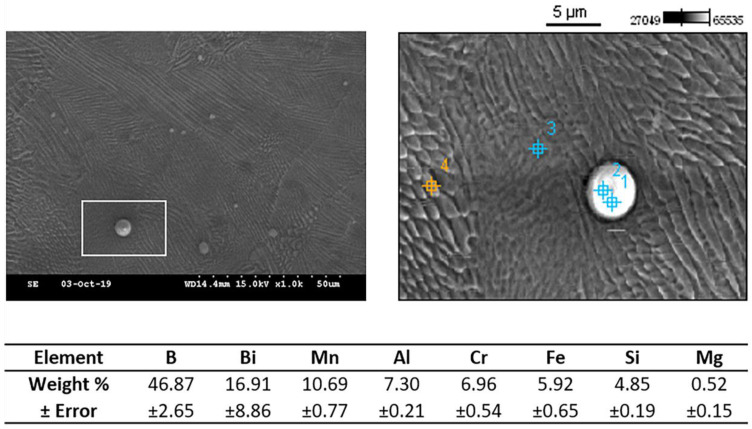
EDX analysis of the residual spherical particles seen after experiment 2B. The average values from locations 1 and 2 were listed in table, while the values from location 3 and 4 were measured for references.

**Figure 17 materials-14-01376-f017:**
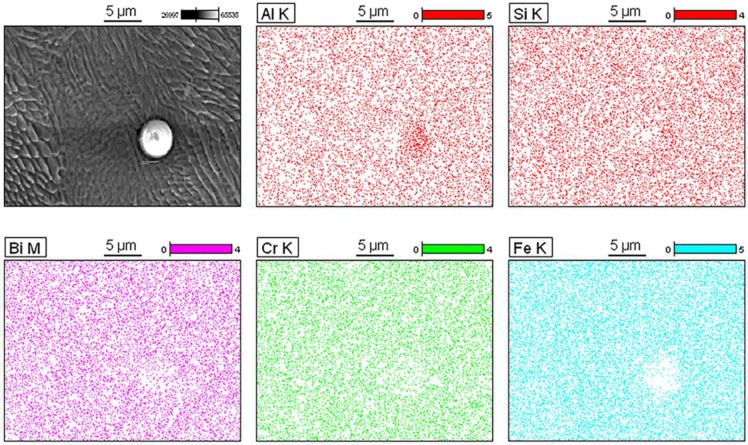
EDX analysis of the residual particles using energy dispersive X-ray mapping with key elements shown.

**Figure 18 materials-14-01376-f018:**
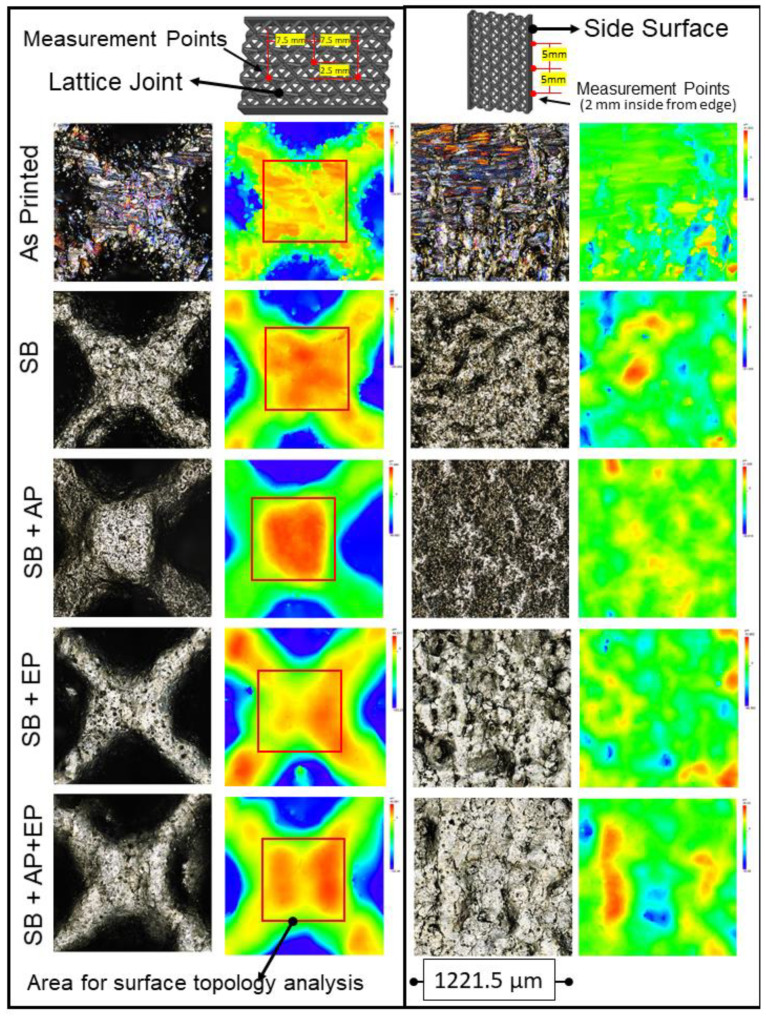
Surface topography measurements following the various combinations of sequential post-processing of the metal lattices. (SB: sandblasting, AP: abrasive polishing, EP: electropolishing.).

**Figure 19 materials-14-01376-f019:**
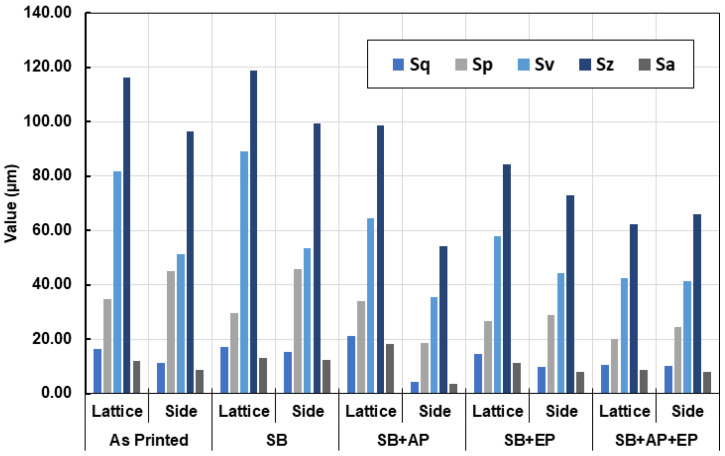
Measured conventional height parameters for the lattice joints and side planar surfaces. (Sq: root mean square height, Sp: maximum peak height, Sv: maximum pit height, Sz: maximum height, Sa: arithmetical mean height, SB: Sandblasting, AP: Abrasive Polishing, EP: Electropolishing.).

**Figure 20 materials-14-01376-f020:**
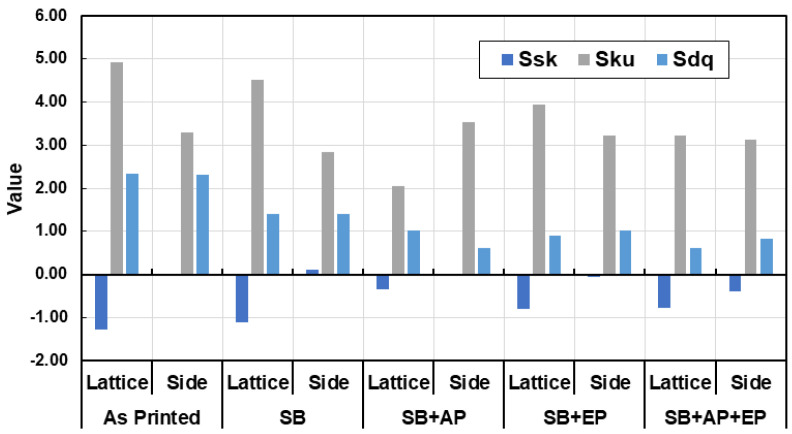
Hybrid parameters measured for the lattice joint and side planar surfaces. (Ssk: skewness, Sku: kurtosis, Sdq: root mean square gradient, SB: sandblasting, AP: abrasive polishing, EP: electropolishing.)

**Figure 21 materials-14-01376-f021:**
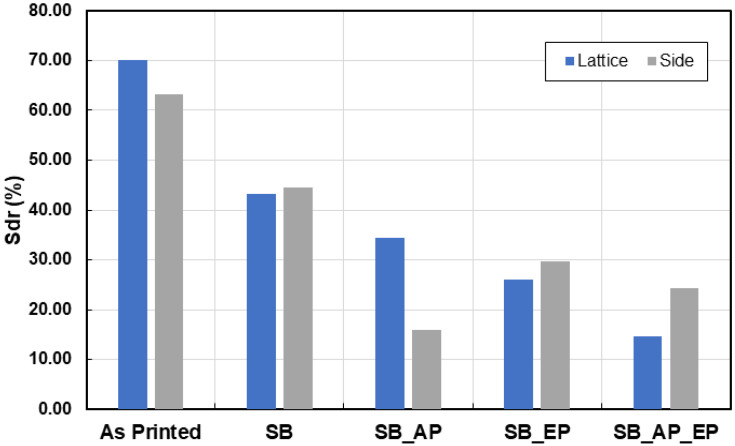
Developed interfacial area ratio (Sdr) measurements for lattice joint and side planar surfaces.

**Figure 22 materials-14-01376-f022:**
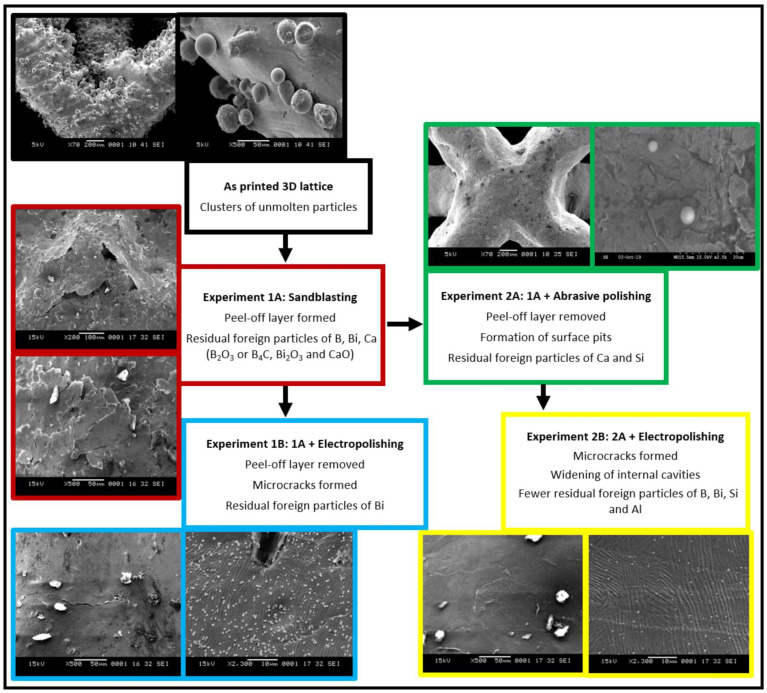
Summary of the effects of post processing treatments on the surface features and foreign particle distributions.

**Table 1 materials-14-01376-t001:** Nominal composition of the stainless steel 316L powder used for lattice production.

Elements	Mass (%)
Iron	Balance
Chromium	16 to 18
Nickel	10 to 14
Molybdenum	2 to 3
Manganese	≤2
Silicon	≤1
Nitrogen	<0.1
Oxygen	≤0.1
Phosphorus	≤0.045
Carbon	≤0.03
Sulphur	≤0.03

**Table 2 materials-14-01376-t002:** Printing parameters used in the selective laser melting process.

Laser power	200 W
Scan speed	600 mm/s
Hatch distance	0.06 mm
Layer thickness	0.05 mm
Laser spot size or focus diameter	0.07 mm
Laser exposure time	50 ms
Powder size	0.025–0.045 mm

## Data Availability

The data presented in this study are available on request from corresponding author.

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
