# Peer review of "Post-Processing and Surface Characterization of Additively Manufactured Stainless Steel 316L Lattice: Implications for BioMedical Use"

_materials, 2021, doi:10.3390/ma14061376_

Round 1
Reviewer 1 Report
Dear authors,
An interesting article on the characterization of the post-processing surface additively manufactured. Why only two roughness parameters were selected for surface evaluation. The Sa parameter does not reflect the nature of the surface. Despite the same value of parameter Sa, the character of the surface texture may be different, while parameter Sz is sensitive to the peaks of the matrix of measurement points. Have the authors considered other roughness parameters e.g. horizontal or hybrid parameters?
Nowadays, for the assessment of surfaces, also manufactured additively, multiscale methods are used, which allow assessing particular features in certain scales. This will allow assessing not only visible on the figure particles but also a complex analysis of such surfaces, evaluation of the influence of different types of processes on the final product. I strongly recommend the multiscale approach. It should be also accompanied with statistical analysis what now is missing.
In addition, these references can improve paper quality. Please try to include that in your study.
T. Kozior, J. Bochnia, P. Zmarzły, D. Gogolewski, T.G. Mathia, Waviness of freeform surface characterizations from austenitic stainless steel (316l) manufactured by 3d printing-selective laser melting (slm) technology, Materials (Basel). 13 (2020). doi:10.3390/ma13194372.
Kind regards
Reviewer
Author Response
Thank you for giving us the opportunity to revise and resubmit our paper in light of the comments made by the reviewers. We have found the comments to be very helpful, and have revised the manuscript accordingly. The changes in the manuscript have been highlighted in yellow.
Please see the attachment of point-by-point responses.

Reviewer 2 Report
Dear Authors,
I have read your paper "Characterisation of the Post-Processing Surface of An Additively Manufactured Stainless Steel 316L Lattice: Concerns for Biomedical Use" carefully.
This paper describes the effect of post-treatment on the roughness of the SLM printed 316L lattice.
The materials and methods are properly described, so that other research groups may reproduce them. Explanations are clear and the paper is easy to read.
For this reason, the paper is interesting and worth of publication. However, it requires few corrections.
1. There is no information about the diameter of the nozzle of the sandblasting system. What is the average distance between the surface and the nozzle?
2. Please, add the proportion of the component in the mixture of hard scrub, sand.
3. Why didn’t you use the ultrasonic cleaning in the liquid for post-processing after sandblasting?
4. Usually, for surface cleaning the powder with the same chemical composition is used. Can you add information about post-processing with blasting with the same powder?
5. Is there change in the chemical composition of the surface layer after electropolishing?
The paper can be accepted for publication only after major improvements.
Author Response
Thank you for giving us the opportunity to revise and resubmit our paper in light of the comments made by the reviewers. We have found the comments to be very helpful, and have revised the manuscript accordingly. The changes in the manuscript have been highlighted in yellow.
Please see the attachment of point-by-point response report.

Reviewer 3 Report
The paper is generally well-written but I have some concerns related to this study:
- It is challenging to capture the complexity of surface topography of additively manufactured metal components, especially of lattice structure. Since you analyze the effect of certain finishing technologies on the final surface texture I would expect to have the metrological side better covered in the introduction. This include measurement and instrumentation (SEM, confocal, microCT, FVM etc.) as well as characterization (feature-based, aspects related to re-entrant features, especially in terms of calculation ISO 25178 standard parameters like Sa, Sq and others). See e.g.: https://doi.org/10.1016/j.measurement.2019.04.027
I would expect at least 10 or more examples from the literature to be included in this paper.
- Establishing functional correlations between surface characterization parameters and technological parameters is crucial in this kind of studies and should be better handled. Some aspects of it are shown in: https://doi.org/10.1016/j.cirp.2018.06.001 . Please do elaborate on that in the introduction.
- A brief explanation of why the presented finishing technologies were used in this study should be given. There are also other finishing technologies like shape adaptive grinding or laser assisted polishing which are not mentioned here. This should be amended.
- Manufacturers of the finishing equipment should be provided.
- As for digital microscopy, more details should be given. What kind of magnification did you use? What kind of optical microscopy it belongs (confocal, WLI, FVM others)? What was the numerical aperture? How did you postprocess the data? Were outliers removed and how? Were non-measured regions filled? What was the cut-off filter wavelength how it was determined? Was form removed? Was roughness and waviness decomposed and what was the filtration used here for that purpose? What was the software you used for calculating Sa and Sq? Some examples measurement via this “digital” microscopy should be provided.
- Generally speaking Sa and Sq are not the most well-suited parameters to capture the complexity of surface topography of additively manufactured parts. Perhaps hybrid (Sdr for sure) and feature parameters would be better – see ISO 25178 for more details. From the potential application point of view functional parameters might be prospective for your analysis too. Anyway, please do expand this section and provide more results to improve the reception of this paper.
- The discussion is extremely superficial and has to be significantly expanded. The novelty of this results should be better explained and discussed with results achieved by other scholars. The topic is new but you are not the first who deal with surface finish of AM metal parts. I would expect at least 1-2 pages for that.
Author Response

(The authors gave the same response as above.)

Round 2
Reviewer 2 Report
Dear Authors,
I have read your modified paper "Characterisation of the Post-processing Surface of An Addi-tively Manufactured Stainless Steel 316L Lattice: Concerns for Biomedical Use" carefully.
The materials and methods are properly described, so that other research groups may reproduce them. Explanations are clear and the paper is easy to read.
I can recommend the Editor to accept this revised manuscript to be published in Materials."
Author Response
Dear Reviewer: Many thanks for your recommendation.
Reviewer 3 Report
Thank you very much for addressing my comments in the revised paper. I have two minor requests:
- Please use "×" instead of "X" to indicate multiplication. This concerns magnification "20X", size of measured area and some more.
- Please provide sampling intervals in X- and Y- direction for the confocal microscopic measurements. Number of points in both direction can also be given in brackets.
Author Response
Thank you very much for addressing my comments in the revised paper. I have two minor requests:
1. Please use "×" instead of "X" to indicate multiplication. This concerns magnification "20X", size of measured area and some more.
Many thanks for the comment, we have checked carefully and removed all the typos.
2. Please provide sampling intervals in X- and Y- direction for the confocal microscopic measurements. Number of points in both direction can also be given in brackets.
We have revised Fig. 18, indicated the sampling intervals and all the measurement points (page 17).
The authors would like to thank the reviewer for the careful reading. We have revised the manuscript thoroughly.